# Synthesizing Robust Plans under Incomplete Domain Models

**Tuan A. Nguyen**
Arizona State University
natuan@asu.edu

**Subbarao Kambhampati**
Arizona State University
rao@asu.edu

**Minh Do**
NASA Ames Research Center
minh.do@nasa.gov

## Abstract

Most current planners assume complete domain models and focus on generating correct plans. Unfortunately, domain modeling is a laborious and error-prone task, thus real world agents have to plan with incomplete domain models. While domain experts cannot guarantee completeness, often they are able to circumscribe the incompleteness of the model by providing annotations as to which parts of the domain model may be incomplete. In such cases, the goal should be to synthesize plans that are robust with respect to any known incompleteness of the domain. In this paper, we first introduce annotations expressing the knowledge of the domain incompleteness and formalize the notion of plan robustness with respect to an incomplete domain model. We then show an approach to compiling the problem of finding robust plans to the conformant probabilistic planning problem, and present experimental results with Probabilistic-FF planner.

## 1   Introduction

In the past several years, significant strides have been made in scaling up plan synthesis techniques. We now have technology to routinely generate plans with hundreds of actions. All this work, however, makes a crucial assumption—that the action models of an agent are completely known in advance. While there are domains where knowledge-engineering such detailed models is necessary and feasible (e.g., mission planning domains in NASA and factory-floor planning), it is increasingly recognized (c.f. [13]) that there are also many scenarios where insistence on correct and complete models renders the current planning technology unusable. The incompleteness in such cases arises because domain writers do not have the full knowledge of the domain physics. One tempting idea is to wait until the models become complete, either by manual revision or by machine learning. Alas, the users often don't have the luxury of delaying their decision making. For example, although there exist efforts [1, 26] that attempt to either learn models from scratch or revise existing ones, their operation is contingent on the availability of successful plan traces, or access to execution experience. There is thus a critical need for planning technology that can get by with partially specified domain models, and yet generate plans that are "robust" in the sense that they are likely to execute successfully in the real world.

This paper addresses the problem of formalizing the notion of plan robustness with respect to an incomplete domain model, and connects the problem of generating a robust plan under such model to *conformant probabilistic planning* [15, 11, 2, 4]. Following Garland & Lesh [7], we shall assume that although the domain modelers cannot provide complete models, often they are able to provide annotations on the partial model circumscribing the places where it is incomplete.In our framework, these annotations consist of allowing actions to have *possible* preconditions and effects (in addition to the standard necessary preconditions and effects).

As an example, consider a variation of the *Gripper* domain, a well-known planning benchmark domain. The robot has one gripper that can be used to pick up balls, which are of two types light and heavy, from one room and move them to another room. The modeler suspects that the gripper may have an internal problem, but this cannot be confirmed until the robot actually executes the plan. If it actually has the problem, the execution of the *pick-up* action succeeds only with balls that are *not*

heavy, but if it has no problem, it can always pickup all types of balls. The modeler can express this partial knowledge about the domain by annotating the action with a statement representing the possible precondition that balls should be light.

Incomplete domain models with such possible preconditions and effects implicitly define an exponential set of complete domain models, with the semantics that the real domain model is guaranteed to be one of these. The robustness of a plan can now be formalized in terms of the cumulative probability mass of the complete domain models under which it succeeds. We propose an approach that compiles the problem of finding robust plans into the conformant probabilistic planning problem. We then present empirical results showing interesting relation between aspects such as the amount domain incompleteness, solving time and plan quality.

## 2 Problem Formulation

We define an *incomplete domain model* $\widetilde{\mathcal{D}}$ as $\widetilde{\mathcal{D}} = \langle F, A \rangle$, where $F = \{p_1, p_2, ..., p_m\}$ is a set of *propositions*, $A$ is a set of *actions* $a$, each might be incompletely specified. We denote $\mathbf{T}$ and $\mathbf{F}$ as the *true* and *false* truth values of propositions. A *state* $s \subseteq F$ is a set of propositions. In addition to proposition sets that are known as its preconditions $Pre(a) \subseteq F$, add effects $Add(a) \subseteq F$ and delete effects $Del(a) \subseteq F$, each action $a \in A$ also contains the following annotations:

- Possible precondition set $\widetilde{Pre}(a) \subseteq F \setminus Pre(a)$ contains propositions that action $a$ *might* need as its preconditions.

- Possible add (delete) effect set $\widetilde{Add}(a) \subseteq F \setminus Add(a)$ ($\widetilde{Del}(a) \subseteq F \setminus Del(a)$) contains propositions that the action $a$ *might* add (delete, respectively) after its execution.

In addition, each possible precondition, add and delete effect $p$ of the action $a$ is associated with a weight $w_a^{pre}(p)$, $w_a^{add}(p)$ and $w_a^{del}(p)$ ($0 < w_a^{pre}(p), w_a^{add}(p), w_a^{del}(p) < 1$) representing the domain writer's assessment of the likelihood that $p$ will actually be *realized* as a precondition, add and delete effect of $a$ (respectively) during plan execution. Possible preconditions and effects whose likelihood of realization is not given are assumed to have weights of $\frac{1}{2}$. Propositions that are not listed in those "possible lists" of an action are assumed to be *not* affecting or being affected by the action.[1]

Given an incomplete domain model $\widetilde{\mathcal{D}}$, we define its *completion set* $\langle\!\langle \widetilde{\mathcal{D}} \rangle\!\rangle$ as the set of *complete* domain models whose actions have all the necessary preconditions, adds and deletes, and a *subset* of the possible preconditions, possible adds and possible deletes. Since any subset of $\widetilde{Pre}(a)$, $\widetilde{Add}(a)$ and $\widetilde{Del}(a)$ can be realized as preconditions and effects of action $a$, there are exponentially large number of possible *complete* domain models $\mathcal{D}_i \in \langle\!\langle \widetilde{\mathcal{D}} \rangle\!\rangle = \{\mathcal{D}_1, \mathcal{D}_2, ..., \mathcal{D}_{2^K}\}$, where $K = \sum_{a \in A}(|\widetilde{Pre}(a)| + |\widetilde{Add}(a)| + |\widetilde{Del}(a)|)$. For each complete model $\mathcal{D}_i$, we denote the corresponding sets of realized preconditions and effects for each action $a$ as $\overline{Pre}_i(a)$, $\overline{Add}_i(a)$ and $\overline{Del}_i(a)$; equivalently, its complete sets of preconditions and effects are $Pre(a) \cup \overline{Pre}_i(a)$, $Add(a) \cup \overline{Add}_i(a)$ and $Del(a) \cup \overline{Del}_i(a)$.

The projection of a sequence of actions $\pi$ from an initial state $I$ according to an incomplete domain model $\widetilde{\mathcal{D}}$ is defined in terms of the projections of $\pi$ from $I$ according to each complete domain model $\mathcal{D}_i \in \langle\!\langle \widetilde{\mathcal{D}} \rangle\!\rangle$:

$$\gamma(\pi, I, \widetilde{\mathcal{D}}) = \{\gamma(\pi, I, \mathcal{D}_i) \mid \mathcal{D}_i \in \langle\!\langle \widetilde{\mathcal{D}} \rangle\!\rangle\} \qquad (1)$$

where the projection over complete models is defined in the usual STRIPS way, with one important difference. Specifically, the result of applying an action $a$, which is complete in $\mathcal{D}_i$, in a state $s$ is defined as followed:

$$\gamma(\langle a \rangle, s, \mathcal{D}_i) = (s \setminus (Del(a) \cup \overline{Del}_i(a))) \cup (Add(a) \cup \overline{Add}_i(a)),$$

if all preconditions of $a$ are satisfied in $s$, and is taken to be $s$ otherwise (rather than as an *undefined* state)—in other words, actions in our setting have "soft" preconditions and thus are applicable in any state. Such a *generous execution semantics* (GES) is critical from an application point of view: With

incomplete models, failure of actions should be expected, and the plan needs to be "robustified" against them during synthesis. The GES facilitates this by ensuring that the plan as a whole does not have to fail if an individual action fails (without it, failing actions doom the plan and thus cannot be supplanted). The resulting state of applying a sequence of complete actions $\pi = \langle a_1, ..., a_n \rangle$ in $s$ with respects to $\mathcal{D}_i$ is defined as:

$$\gamma(\pi, s, \mathcal{D}_i) = \gamma(\langle a_n \rangle, \gamma(\langle a_1, ..., a_{n-1} \rangle, s, \mathcal{D}_i), \mathcal{D}_i).$$

A *planning problem* with incomplete domain $\widetilde{\mathcal{D}}$ is $\widetilde{\mathcal{P}} = \langle \widetilde{\mathcal{D}}, I, G \rangle$ where $I \subseteq F$ is the set of propositions that are true in the *initial state* (and all the remaining are false), and $G$ is the set of *goal propositions*. An action sequence $\pi$ is considered a *valid* plan for $\widetilde{\mathcal{P}}$ if $\pi$ solves the problem in at least one completion of $\langle\langle \widetilde{\mathcal{D}} \rangle\rangle$. Specifically, $\exists_{\mathcal{D}_i \in \langle\langle \widetilde{\mathcal{D}} \rangle\rangle} \gamma(\pi, I, \mathcal{D}_i) \models G$. Given that $\langle\langle \widetilde{\mathcal{D}} \rangle\rangle$ can be exponentially large in terms of possible preconditions and effects, validity is too weak to guarantee on the quality of the plan. What we need is a notion that $\pi$ succeeds in most of the highly likely completions of $\widetilde{\mathcal{D}}$. We do this in terms of a *robustness* measure, which will be presented in the next section.

**Modeling assumptions underlying our formulation:** From the modeling point of view, the possible precondition and effect sets can be modeled at either the grounded action or action schema level (and thus applicable to all grounded actions sharing the same action schema). From a practical point of view, however, incompleteness annotations at ground level hugely increase the burden on domain writers. In our formal treatment, we therefore assume that annotations are specified at the schema level.

```
pick-up
:parameters (?b - ball ?r - room)
:precondition
    (and (at ?b ?r) (at-robot ?r) (free-gripper))
:possible_precondition
    (and (light ?b))
:effect
    (and (carry ?b)(not (at ?b ?r))(not (free-gripper)))
:possible_effect
    (and (dirty ?b))
```

Figure 1: Decription of incomplete action schema *pick-up* in *Gripper* domain.

Since possible preconditions and effects can be represented as random variables, they can in principle be modeled using graphical models such as Makov Logic Networks and Bayesian Networks [14]. Though it appears to be an interesting technical challenge, this would require a significant additional knowledge input from the domain writer, and thus less likely to be helpful in practice. We therefore assume that the possible preconditions and effects are uncorrelated, thus can be realized independently (both within each action schema and across different ones).

**Example:** Figure 1 shows the description of incomplete action *pick-up(?b - ball,?r - room)* as described above at the schema level. In addition to the possible precondition *(light ?b)* on the weight of the ball *?b*, we also assume that since the modeler is unsure if the gripper has been cleaned or not, she models it with a possible add effect *(dirty ?b)* indicating that the action might make the ball dirty. Those two possible preconditions and effects can be realized independently, resulting in four possible candidate complete domains (assuming all other action schemas in the domain are completely described).

## 3   A Robustness Measure for Plans

The robustness of a plan $\pi$ for the problem $\widetilde{\mathcal{P}} = \langle \widetilde{\mathcal{D}}, I, G \rangle$ is defined as the cumulative probability mass of the completions of $\widetilde{\mathcal{D}}$ under which $\pi$ succeeds (in achieving the goals). More formally, let $\mathbf{Pr}(\mathcal{D}_i)$ be the probability distribution representing the modeler's estimate of the probability that a given model in $\langle\langle \widetilde{\mathcal{D}} \rangle\rangle$ is the real model of the world (such that $\sum_{\mathcal{D}_i \in \langle\langle \widetilde{\mathcal{D}} \rangle\rangle} \mathbf{Pr}(\mathcal{D}_i) = 1$). The robustness of $\pi$ is defined as follows:

$$R(\pi, \widetilde{\mathcal{P}} : \langle \widetilde{\mathcal{D}}, I, G \rangle) \stackrel{def}{\equiv} \sum_{\mathcal{D}_i \in \langle\langle \widetilde{\mathcal{D}} \rangle\rangle, \gamma(\pi, I, \mathcal{D}_i) \models G} \mathbf{Pr}(\mathcal{D}_i) \qquad (2)$$

It is easy to see that if $R(\pi, \widetilde{\mathcal{P}}) > 0$, then $\pi$ is a valid plan for $\widetilde{\mathcal{P}}$.

Note that given the uncorrelated incompleteness assumption, the probability $\mathbf{Pr}(\mathcal{D}_i)$ for a model $\mathcal{D}_i \in \langle\langle \widetilde{\mathcal{D}} \rangle\rangle$ can be computed as the product of the weights $w_a^{pre}(p)$, $w_a^{add}(p)$, and $w_a^{del}(p)$ for all $a \in A$ and its possible preconditions/effects $p$ if $p$ *is* realized in the model (or the product of their "complement" $1 - w_a^{pre}(p)$, $1 - w_a^{add}(p)$, and $1 - w_a^{del}(p)$ if $p$ is *not* realized).

**Example:** Figure 2 shows an example with an incomplete domain model $\widetilde{\mathcal{D}} = \langle F, A \rangle$ with $F = \{p_1, p_2, p_3\}$ and $A = \{a_1, a_2\}$ and a solution plan $\pi = \langle a_1, a_2 \rangle$ for the problem $\widetilde{\mathcal{P}} = \langle \widetilde{\mathcal{D}}, I = \{p_2\}, G = \{p_3\} \rangle$. The incomplete model is: $Pre(a_1) = \emptyset$, $\widetilde{Pre}(a_1) = \{p_1\}$, $Add(a_1) = \{p_2, p_3\}$, $\widetilde{Add}(a_1) = \emptyset$, $Del(a_1) = \emptyset$, $\widetilde{Del}(a_1) = \emptyset$; $Pre(a_2) = \{p_2\}$, $\widetilde{Pre}(a_2) = \emptyset$, $Add(a_2) = \emptyset$, $\widetilde{Add}(a_2) = \{p_3\}$, $Del(a_2) = \emptyset$, $\widetilde{Del}(a_2) = \{p_1\}$. Given that the total number of possible preconditions and effects is 3, the total number of completions ($|\langle\langle\widetilde{\mathcal{D}}\rangle\rangle|$) is $2^3 = 8$, for each of which the plan $\pi$ may succeed or fail to achieve $G$, as shown in the table. In the fifth candidate model, for instance, $p_1$ and $p_3$ are realized as precondition and add effect of $a_1$ and $a_2$, whereas $p_1$ is not a delete effect of action $a_2$. Even though $a_1$ could not execute (and thus $p_3$ remains *false* in

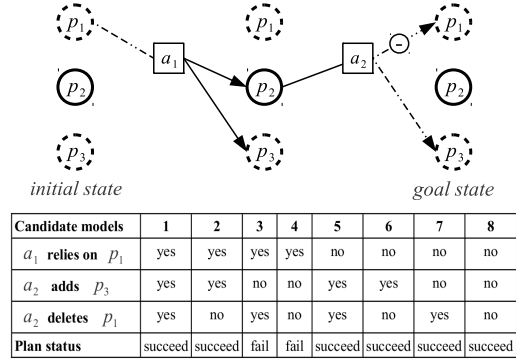

| Candidate models | 1 | 2 | 3 | 4 | 5 | 6 | 7 | 8 |
|---|---|---|---|---|---|---|---|---|
| $a_1$ **relies on** $p_1$ | yes | yes | yes | yes | no | no | no | no |
| $a_2$ **adds** $p_3$ | yes | yes | no | no | yes | yes | no | no |
| $a_2$ **deletes** $p_1$ | yes | no | yes | no | yes | no | yes | no |
| **Plan status** | succeed | succeed | fail | fail | succeed | succeed | succeed | succeed |

Figure 2: Example for a set of complete candidate domain models, and the corresponding plan status. Circles with solid and dash boundary respectively are propositions that are known to be **T** and might be **F** when the plan executes (see more in text).

the second state), the goal eventually is achieved by action $a_2$ with respects to this candidate model. Overall, there are two of eight candidate models where $\pi$ fails and six for which it succeeds. The robustness value of the plan is $R(\pi) = \frac{3}{4}$ if $\mathbf{Pr}(\mathcal{D}_i)$ is the uniform distribution. However, if the domain writer thinks that $p_1$ is very likely to be a precondition of $a_1$ and provides $w_{a_1}^{pre}(p_1) = 0.9$, the robustness of $\pi$ decreases to $R(\pi) = 2 \times (0.9 \times 0.5 \times 0.5) + 4 \times (0.1 \times 0.5 \times 0.5) = 0.55$ (as intuitively, the last four models with which $\pi$ succeeds are very unlikely to be the real one). Note that under the standard non-generous execution semantics (non-GES) where action failure causes plan failure, the plan $\pi$ would be mistakenly considered failing to achieve $G$ in the first two complete models, since $a_2$ is prevented from execution.

### 3.1 A Spectrum of Robust Planning Problems

Given this set up, we can now talk about a spectrum of problems related to planning under incomplete domain models:

**Robustness Assessment (RA):** Given a plan $\pi$ for the problem $\widetilde{\mathcal{P}}$, assess the robustness of $\pi$.

**Maximally Robust Plan Generation (RG\*):** Given a problem $\widetilde{\mathcal{P}}$, generate the maximally robust plan $\pi^*$.

**Generating Plan with Desired Level of Robustness (RG$^\rho$):** Given a problem $\widetilde{\mathcal{P}}$ and a robustness threshold $\rho$ ($0 < \rho \leq 1$), generate a plan $\pi$ with robustness greater than or equal to $\rho$.

**Cost-sensitive Robust Plan Generation (RG$_c^*$):** Given a problem $\widetilde{\mathcal{P}}$ and a cost bound $c$, generate a plan $\pi$ of maximal robustness subject to cost bound $c$ (where the cost of a plan $\pi$ is defined as the cumulative costs of the actions in $\pi$).

**Incremental Robustification (RI$_c$):** Given a plan $\pi$ for the problem $\widetilde{\mathcal{P}}$, improve the robustness of $\pi$, subject to a cost budget $c$.

The problem of assessing robustness of plans, RA, can be tackled by compiling it into a weighted model-counting problem. The following theorem shows that RA with uniform distribution of candidate complete models is complete for $\#P$ complexity class [22], and thus the robustness assessment problem is at least as hard as NP-complete.[2]

**Theorem 1.** *The problem of assessing plan robustness with the uniform distribution of candidate complete models is $\#P$-complete.*

For plan synthesis problems, we can talk about either generating a maximally robust plan, RG\*, or finding a plan with a robustness value above the given threshold, RG$^\rho$. A related issue is that of the

interaction between plan cost and robustness. Often, increasing robustness involves using additional (or costlier) actions to support the desired goals, and thus comes at the expense of increased plan cost. We can also talk about cost-constrained robust plan generation problem $RG_c^*$. Finally, in practice, we are often interested in increasing the robustness of a given plan (either during iterative search, or during mixed-initiative planning). We thus also have the incremental variant $RI_c$. In the next section, we will focus on the problem of synthesizing plans with at least a robustness value $\rho$.

# 4 Synthesizing Robust Plans

Given a planning problem $\widetilde{\mathcal{P}}$ with an incomplete domain $\widetilde{\mathcal{D}}$, the ultimate goal is to synthesize a plan having a desired level of robustness, or one with maximal robustness value. In this section, we will show that the problem of generating plan with at least $\rho$ robustness ($0 < \rho \leq 1$), can be compiled into an equivalent conformant probabilistic planning problem. The most robust plan can then be found with a sequence of increasing threshold values.

## 4.1 Conformant Probabilistic Planning

Following the formalism in [4], a domain in conformant probabilistic planning (CPP) is a tuple $\mathcal{D}' = \langle F', A' \rangle$, where $F'$ and $A'$ are the sets of propositions and probabilistic actions, respectively. A belief state $b : 2^{F'} \to [0, 1]$ is a distribution of states $s \subseteq F'$ (we denote $s \in b$ if $b(s) > 0$). Each action $a' \in A'$ is specified by a set of preconditions $Pre(a') \subseteq F'$ and conditional effects $E(a')$. For each $e = (cons(e), \mathcal{O}(e)) \in E(a')$, $cons(e) \subseteq F'$ is the condition set and $\mathcal{O}(e)$ determines the set of outcomes $\varepsilon = (Pr(\varepsilon), add(\varepsilon), del(\varepsilon))$ that will add and delete proposition sets $add(\varepsilon), del(\varepsilon)$ into and from the resulting state with the probability $Pr(\varepsilon)$ ($0 \leq Pr(\varepsilon) \leq 1$, $\sum_{\varepsilon \in \mathcal{O}(e)} Pr(\varepsilon) = 1$). All condition sets of the effects in $E(a')$ are assumed to be mutually exclusive and exhaustive. The action $a'$ is applicable in a belief state $b$ if $Pre(a') \subseteq s$ for all $s \in b$, and the probability of a state $s'$ in the resulting belief state is $b_{a'}(s') = \sum_{s \supseteq Pre(a')} b(s) \sum_{\varepsilon \in \mathcal{O}'(e)} Pr(\varepsilon)$, where $e \in E(a')$ is the conditional effect such that $cons(e) \subseteq s$, and $\mathcal{O}'(e) \subseteq \mathcal{O}(e)$ is the set of outcomes $\varepsilon$ such that $s' = s \cup add(\varepsilon) \setminus del(\varepsilon)$.

Given the domain $\mathcal{D}'$, a problem $\mathcal{P}'$ is a quadruple $\mathcal{P}' = \langle \mathcal{D}', b_I, G', \rho' \rangle$, where $b_I$ is an initial belief state, $G'$ is a set of goal propositions and $\rho'$ is the acceptable goal satisfaction probability. A sequence of actions $\pi' = (a'_1, ..., a'_n)$ is a solution plan for $\mathcal{P}'$ if $a'_i$ is applicable in the belief state $b_i$ (assuming $b_1 \equiv b_I$), which results in $b_{i+1}$ ($1 \leq i \leq n$), and it achieves all goal propositions with at least $\rho'$ probability.

## 4.2 Compilation

Given an incomplete domain model $\widetilde{\mathcal{D}} = \langle F, A \rangle$ and a planning problem $\widetilde{\mathcal{P}} = \langle \widetilde{\mathcal{D}}, I, G \rangle$, we now describe a compilation that translates the problem of synthesizing a solution plan $\pi$ for $\widetilde{\mathcal{P}}$ such that $R(\pi, \widetilde{\mathcal{P}}) \geq \rho$ to a CPP problem $\mathcal{P}'$. At a high level, the realization of possible preconditions $p \in \widetilde{Pre}(a)$ and effects $q \in \widetilde{Add}(a)$, $r \in \widetilde{Del}(a)$ of an action $a \in A$ can be understood as being determined by the truth values of *hidden* propositions $p_a^{pre}$, $q_a^{add}$ and $r_a^{del}$ that are certain (i.e. unchanged in any world state) but unknown. Specifically, the applicability of the action in a state $s \subseteq F$ depends on possible preconditions $p$ that are realized (i.e. $p_a^{pre} = \mathbf{T}$), and their truth values in $s$. Similarly, the values of $q$ and $r$ are affected by $a$ in the resulting state only if they are realized as add and delete effects of the action (i.e., $q_a^{add} = \mathbf{T}$, $r_a^{del} = \mathbf{T}$). There are totally $2^{|\widetilde{Pre}(a)| + |\widetilde{Add}(a)| + |\widetilde{Del}(a)|}$ realizations of the action $a$, and all of them should be considered simultaneously in checking the applicability of the action and in defining corresponding resulting states.

With those observations, we use multiple conditional effects to compile away incomplete knowledge on preconditions and effects of the action $a$. Each conditional effect corresponds to one realization of the action, and can be fired only if $p = \mathbf{T}$ whenever $p_a^{pre} = \mathbf{T}$, and adding (removing) an effect $q$ ($r$) into (from) the resulting state depending on the values of $q_a^{add}$ ($r_a^{del}$, respectively) in the realization.

While the partial knowledge can be removed, the hidden propositions introduce uncertainty into the initial state, and therefore making it a *belief* state. Since actions are always applicable in our formulation, resulting in either a new or the same successor state, preconditions $Pre(a)$ must be modeled as conditions of all conditional effects. We are now ready to formally specify the resulting domain $\mathcal{D}'$ and problem $\mathcal{P}'$.

For each action $a \in A$, we introduce new propositions $p_a^{pre}$, $q_a^{add}$, $r_a^{del}$ and their negations $np_a^{pre}$, $nq_a^{add}$, $nr_a^{del}$ for each $p \in \widetilde{Pre}(a)$, $q \in \widetilde{Add}(a)$ and $r \in \widetilde{Del}(a)$ to determine whether they are realized as preconditions and effects of $a$ in the real domain.[3] Let $F_{new}$ be the set of those new propositions, then $F' = F \cup F_{new}$ is the proposition set of $\mathcal{D}'$.

Each action $a' \in A'$ is made from one action $a \in A$ such that $Pre(a') = \emptyset$, and $E(a')$ consists of $2^{|\widetilde{Pre}(a)|+|\widetilde{Add}(a)|+|\widetilde{Del}(a)|}$ conditional effects $e$. For each conditional effect $e$:

- $cons(e)$ is the union of the following sets: (i) the certain preconditions $Pre(a)$, (ii) the set of possible preconditions of $a$ that are realized, and hidden propositions representing their realization: $\overline{Pre}(a) \cup \{p_a^{pre}|p \in \overline{Pre}(a)\} \cup \{np_a^{pre}|p \in \widetilde{Pre}(a) \setminus \overline{Pre}(a)\}$, (iii) the set of hidden propositions corresponding to the realization of possible add (delete) effects of $a$: $\{q_a^{add}|q \in \overline{Add}(a)\} \cup \{nq_a^{add}|q \in \widetilde{Add}(a) \setminus \overline{Add}(a)\}$ ($\{r_a^{del}|r \in \overline{Del}(a)\} \cup \{nr_a^{del}|r \in \widetilde{Del}(a) \setminus \overline{Del}(a)\}$, respectively);
- the *single* outcome $\varepsilon$ of $e$ is defined as $add(\varepsilon) = Add(a) \cup \overline{Add}(a)$, $del(\varepsilon) = Del(a) \cup \overline{Del}(a)$, and $Pr(\varepsilon) = 1$,

where $\overline{Pre}(a) \subseteq \widetilde{Pre}(a)$, $\overline{Add}(a) \subseteq \widetilde{Add}(a)$ and $\overline{Del}(a) \subseteq \widetilde{Del}(a)$ represent the sets of realized preconditions and effects of the action. In other words, we create a conditional effect for each subset of the union of the possible precondition and effect sets of the action $a$. Note that the inclusion of new propositions derived from $\overline{Pre}(a)$, $\overline{Add}(a)$, $\overline{Del}(a)$ and their "complement" sets $\widetilde{Pre}(a) \setminus \overline{Pre}(a)$, $\widetilde{Add}(a) \setminus \overline{Add}(a)$, $\widetilde{Del}(a) \setminus \overline{Del}(a)$ makes all condition sets of the action $a'$ mutually exclusive. As for other cases (including those in which some precondition in $Pre(a)$ is excluded), the action has no effect on the resulting state, they can be ignored. The condition sets, therefore, are also exhaustive.

The initial belief state $b_I$ consists of $2^{|F_{new}|}$ states $s' \subseteq F'$ such that $p \in s'$ iff $p \in I$ ($\forall p \in F$), each represents a complete domain model $\mathcal{D}_i \in \langle\!\langle \widetilde{\mathcal{D}} \rangle\!\rangle$ and with the probability $\mathbf{Pr}(\mathcal{D}_i)$, as defined in Section 3. The specification of $b_I$ includes simple Bayesian networks representing the relation between variables in $F_{new}$, e.g. $p_a^{pre}$ and $np_a^{pre}$, where the weights $w(\cdot)$ and $1 - w(\cdot)$ are used to define conditional probability tables. The goal is $G' = G$, and the acceptable goal satisfaction probability is $\rho' = \rho$. Theorem 2 shows the correctness of our compilation. It also shows that a plan for $\widetilde{\mathcal{P}}$ with at least $\rho$ robustness can be obtained directly from solutions of the compiled problem $\mathcal{P}'$.

**Theorem 2.** *Given a plan* $\pi = (a_1, ..., a_n)$ *for the problem* $\widetilde{\mathcal{P}}$, *and* $\pi' = (a_1', ..., a_n')$ *where* $a_k'$ *is the compiled version of* $a_k$ ($1 \le k \le n$) *in* $\mathcal{P}'$. *Then* $R(\pi, \widetilde{\mathcal{P}}) \ge \rho$ *iff* $\pi'$ *achieves all goals with at least* $\rho$ *probability in* $\mathcal{P}'$.

### 4.3 Experimental Results

In this section, we discuss the results of the compilation with Probabilistic-FF (PFF) on variants of the Logistics and Satellite domains, where domain incompleteness is modeled on the preconditions and effects of actions (respectively). Our purpose here is to observe and explain how plan length and synthesizing time vary with the amount of domain incompleteness and the robustness threshold.[4]

*Logistics*: In this domain, each of the two cities $C_1$ and $C_2$ has an airport and a downtown area. The transportation between the two distant cities can only be done by two airplanes $A_1$ and $A_2$. In the downtown area of $C_i$ ($i \in \{1, 2\}$), there are three *heavy* containers $P_{i1}, ..., P_{i3}$ that can be moved to the airport by a truck $T_i$. Loading those containers onto the truck in the city $C_i$, however, requires moving a team of $m$ robots $R_{i1}, ..., R_{im}$ ($m \ge 1$), initially located in the airport, to the downtown area. The source of incompleteness in this domain comes from the assumption that each pair of robots $R_{1j}$ and $R_{2j}$ ($1 \le j \le m$) are made by the same manufacturer $M_j$, both therefore might fail to load a *heavy* container.[5] The actions loading containers onto trucks using robots made

by a particular manufacturer (e.g., the action schema *load-truck-with-robots-of-M1* using robots of manufacturer $M_1$), therefore, have a *possible precondition* requiring that containers should not be heavy. To simplify discussion (see below), we assume that robots of different manufacturers may fail to load heavy containers, though independently, with the same probability of $0.7$. The goal is to transport all three containers in the city $C_1$ to $C_2$, and vice versa. For this domain, a plan to ship a container to another city involves a step of loading it onto the truck, which can be done by a robot (after moving it from the airport to the downtown). Plans can be made more robust by using additional robots of *different* manufacturer after moving them into the downtown areas, with the cost of increasing plan length.

*Satellite*: In this domain, there are two satellites $S_1$ and $S_2$ orbiting the planet Earth, on each of which there are $m$ instruments $L_{i1}, ..., L_{im}$ ($i \in \{1, 2\}$, $m \geq 1$) used to take images of interested modes at some direction in the space. For each $j \in \{1, ..., m\}$, the lenses of instruments $L_{ij}$'s were made from a type of material $M_j$, which might have an error affecting the quality of images that they take. If the material $M_j$ actually has error,

| $\rho$ | $m=1$ | $m=2$ | $m=3$ | $m=4$ | $m=5$ |
|---|---|---|---|---|---|
| 0.1 | 32/10.9 | 36/26.2 | 40/57.8 | 44/121.8 | 48/245.6 |
| 0.2 | 32/10.9 | 36/25.9 | 40/57.8 | 44/121.8 | 48/245.6 |
| 0.3 | 32/10.9 | 36/26.2 | 40/57.7 | 44/122.2 | 48/245.6 |
| 0.4 | $\bot$ | 42/42.1 | 50/107.9 | 58/252.8 | 66/551.4 |
| 0.5 | $\bot$ | 42/42.0 | 50/107.9 | 58/253.1 | 66/551.1 |
| 0.6 | $\bot$ | $\bot$ | 50/108.2 | 58/252.8 | 66/551.1 |
| 0.7 | $\bot$ | $\bot$ | $\bot$ | 58/253.1 | 66/551.6 |
| 0.8 | $\bot$ | $\bot$ | $\bot$ | $\bot$ | 66/550.9 |
| 0.9 | $\bot$ | $\bot$ | $\bot$ | $\bot$ | $\bot$ |

Figure 3: The results of generating robust plans in Logistics domain.

all instruments $L_{ij}$'s produce mangled images. The knowledge of this incompleteness is modeled as a *possible add effect* of the action taking images using instruments made from $M_j$ (for instance, the action schema *take-image-with-instruments-M1* using instruments of type $M_1$) with a probability of $p_j$, asserting that images taken might be in a bad condition. A typical plan to take an image using an instrument, e.g. $L_{14}$ of type $M_4$ on the satellite $S_1$, is first to switch on $L_{14}$, turning the satellite $S_1$ to a ground direction from which $L_{14}$ can be calibrated, and then taking image. Plans can be made more robust by using additional instruments, which might be on a different satellite, but should be of *different* type of materials and can also take an image of the interested mode at the same direction.

Table 3 and 4 shows respectively the results in the Logistics and Satellite domains with $\rho \in \{0.1, 0.2, ..., 0.9\}$ and $m = \{1, 2, ..., 5\}$. The number of complete domain models in the two domains is $2^m$. For Satellite domain, the probabilities $p_j$'s range from $0.25, 0.3, ...$ to $0.45$ when $m$ increases from $1, 2, ...$ to $5$. For each specific value of $\rho$ and $m$, we report $l/t$ where $l$ is the length of plan and $t$ is the running time (in seconds).

| $\rho$ | $m=1$ | $m=2$ | $m=3$ | $m=4$ | $m=5$ |
|---|---|---|---|---|---|
| 0.1 | 10/0.1 | 10/0.1 | 10/0.2 | 10/0.2 | 10/0.2 |
| 0.2 | 10/0.1 | 10/0.1 | 10/0.1 | 10/0.2 | 10/0.2 |
| 0.3 | $\bot$ | 10/0.1 | 10/0.1 | 10/0.2 | 10/0.2 |
| 0.4 | $\bot$ | 37/17.7 | 37/25.1 | 10/0.2 | 10/0.3 |
| 0.5 | $\bot$ | $\bot$ | 37/25.5 | 37/79.2 | 37/199.2 |
| 0.6 | $\bot$ | $\bot$ | 53/216.7 | 37/94.1 | 37/216.7 |
| 0.7 | $\bot$ | $\bot$ | $\bot$ | 53/462.0 | – |
| 0.8 | $\bot$ | $\bot$ | $\bot$ | $\bot$ | – |
| 0.9 | $\bot$ | $\bot$ | $\bot$ | $\bot$ | $\bot$ |

Figure 4: The results of generating robust plans in Satellite domain.

Cases in which no plan is found within the time limit are denoted by "–", and those where it is provable that no plan with the desired robustness exists are denoted by "$\bot$".

As the results indicate, for a fixed amount of domain incompleteness (represented by $m$), the solution plans in both domains tend to be longer with higher robustness threshold $\rho$, and the time to synthesize plans also increases. For instance, in Logistics with $m = 5$, the plan returned has $48$ actions if $\rho = 0.3$, whereas $66$-length plan is needed if $\rho$ increases to $0.4$. On the other hand, we also note that more than the needed number of actions have been used in many solution plans. In the Logistics domain, specifically, it is easy to see that the probability of successfully loading a container onto a truck using robots of $k$ ($1 \leq k \leq m$) different manufacturers is $(1 - 0.7^k)$. However, robots of all five manufacturers are used in a plan when $\rho = 0.4$, whereas using those of three manufacturers is enough. The relaxation employed by PFF that ignores all but one condition in effects of actions, while enables an upper-bound computation for plan robustness, is probably too strong and causes unnecessary increasing in plan length.

Also as we would expect, when the amount of domain incompleteness (i.e., $m$) increases, it takes longer time to synthesize plans satisfying a fixed robustness value $\rho$. As an example, in the Satellite domain, with $\rho = 0.6$ it takes $216.7$ seconds to synthesize a $37$-length plan when $m = 5$, whereas it is only $94.1$ seconds for $m = 4$. Two exceptions can be seen with $\rho = 0.7$ where no plan is found

within the time limit when $m = 5$, although a plan with robustness of $0.7075$ exists in the solution space. A probable explanation for this performance is the costly satisfiability tests and weighted model-counting for computing resulting belief states during the search.

## 5 Related Work

There are currently very few research efforts in automated planning literature that explicitly consider incompletely specified domain models. To our best knowledge, Garland and Lesh [7] were the first discussing incomplete actions and generating robust plans under incomplete domain models. Their notion of plan robustness, however, only has tenuous heuristic connections with the likelihood of successful execution of plans. Weber and Bryce [24] consider a model similar to ours but assume a non-GES formulation during plan synthesis—the plan fails if any action's preconditions are not satisfied. As we mention earlier, this semantics is significantly less helpful from an application point of view; and it is arguably easier. Indeed, their method for generating robust plans relies on the propagation of "reasons" for failure of each action, *assuming that every action before it successfully executes*. Such a propagation is no longer appliable for GES. Morwood and Bryce [16] studied the problem of robustness assessment for the same incompleteness formulation in temporal planning domains, where plan robustness is defined as the number of complete models under which temporal constraints are consistent. The work by Fox et al [6] also explores robustness of plans, but their focus is on temporal plans under unforeseen execution-time variations rather than on incompletely specified domains. Eiter et al [5] introduces language $\mathcal{K}$ for planning under incomplete knowledge. Their formulation is however different from ours in the type of incompleteness (world states v.s. action models) and the notion of plans (secure/conformant plans v.s. robust plans). Our work can also be categorized as one particular instance of the general model-lite planning problem, as defined in [13], in which the author points out a large class of applications where handling incomplete models is unavoidable due to the difficulty in getting a complete model.

As mentioned earlier, there were complementary approaches (c.f. [1, 26]) that attempt to either learn models from scratch or revise existing ones, given the access to successful plan traces or execution experience, which can then be used to solve new planning problems. These works are different from ours in both the additional knowledge about the incomplete model (execution experience v.s. incompleteness annotations), and the notion of solutions (correct with respect to the learned model v.s. to candidate complete models).

Though not directly addressing formulation like ours, the work on *k-fault* plans for non-deterministic planning [12] focused on reducing the "faults" in plan execution. It is however based on the context of stochastic/non-deterministic actions rather than incompletely specified ones. The semantics of the possible preconditions/effects in our incomplete domain models fundamentally differs from non-deterministic and stochastic effects (c.f. work by Kushmerick et al [15]). While the probability of success can be increased by continously executing actions with stochastic effects, the consequence of unknown but deterministic effects is consistent over different executions.

In Markov Decision Processes (MDPs), a fairly rich body of work has been done for *imprecise* transition probabilities [19, 25, 8, 17, 3, 21], using various ways to represent imprecision/incompleteness in the transition models. These works mainly seek for max-min or min-max optimal policies, assuming that Nature acts optimally against the agent. Much of these work is however done at atomic level while we focus on factored planning models. Our incompleteness formulation can also be extended for agent modeling, a topic of interest in multi-agent systems (c.f. [10, 9, 20, 18]).

## 6 Conclusion and Future Work

In this paper, we motivated the need for synthesizing robust plans under incomplete domain models. We introduced annotations for expressing domain incompleteness, formalized the notion of plan robustness, and showed an approach to compile the problem of generating robust plans into conformant probabilistic planning. We presented empirical results showing interesting relation between aspects such as the amount of domain incompleteness, solving time and plan quality. We are working on a direct approach reasoning on correctness constraints of plan prefixes and partial relaxed plans, constrasting it with our compilation method. We also plan to take successful plan traces as a second type of additional inputs for generating robust plans.

**Acknowledgements:** This research is supported in part by the ARO grant W911NF-13-1-0023, the ONR grants N00014-13-1-0176, N00014-09-1-0017 and N00014-07-1-1049, and the NSF grant IIS201330813.

## Footnotes

[1]Our incompleteness annotations therefore can also be used to model domains in which the domain writer can only provide lists of known preconditions/effects of actions, and optionally specifying those known to be *not* in the lists.

[2]The proof is based on a counting reduction from the problem of counting satisfying assignments for MONOTONE-2SAT [23]. We omit it due to the space limit.

[3]These propositions are introduced once, and re-used for all actions sharing the same schema with $a$.

[4]The experiments were conducted using an Intel Core2 Duo 3.16GHz machine with 4Gb of RAM, and the time limit is 15 minutes.

[5]The *uncorrelated incompleteness* assumption applies for possible preconditions of action schemas specified for different manufacturers. It should not be confused here that robots $R_{1j}$ and $R_{2j}$ of the same manufacturer $M_j$ can independently have fault.

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
