[Reviews · NeurIPS 2013]

Submitted by Assigned_Reviewer_5

This paper presented a method to enable the generation of robust plans with partially specified domain models. The motivation of this research topic is well stated. The main contribution of this work is the formalization of the notion of plan robustness with respect to an incomplete domain model. The paper is clearly written and should the general interest for the broad NIPS audience. Here are some comments in hope to help improve the paper in the revision.

The authors nicely listed a spectrum of problems related to robust planning under incomplete domain models. But the main focus in the remaining paper was about the robustness assessment (RA) problem, without detailed discussions on the other problems. This way makes a paper appear to be a bit disconnected. If I understand it correctly, the simulation results are contingent on the weights associated with actions. It is unclear to me how the weights are defined or learned. It would be good to clarify this issue in the paper.
In the simulation result section, it would be nice to show the results from robust planning with complete domain model.
Minor point
Add a brief description about the meaning of numbers in the caption of the two result tables.
Summary: This is a solid paper to address an interesting problem. The paper is clearly written and should the general interest for the broad NIPS audience.

Submitted by Assigned_Reviewer_7

This paper considers the problem of planning under incomplete domain models. The authors formalize domain incompleteness, discuss how to generate robust plans using conformant probabilistic planning, and compare robustness to solving time and plan quality in two test domains.

Clarity: The paper is very well written and the authors do a great job at explaining all of the details both formally and informally. Although I am mostly unfamiliar with planning research, I was able to follow along fairly well and understand the contributions. Organization is good and the example at the top of page 4 is great, but I believe that on line 177, the "fifth candidate model" described is actually the second candidate model given by the table in Figure 2.

Quality: What I like most about this paper is that the incompleteness formulation is very intuitive and easy to understand. A practioner should be able to read the paper and integrate the ideas into their own planning problem without too much worry. Theorems of completeness and correctness are also stated. However, perhaps the biggest drawback is the lack of proofs, particularly for Theorem 1. While I agree with the authors that there is no space in the main body of the paper to include proofs, the authors really should have included them as supplementary material.

Originality: While there are some similarities here to other work, the authors do discuss this in the related work section and I believe that the proposed formulation for incompleteness is adequately original and worthy of publication. The authors clearly state how their work differs from previous contributions.

Significance: I believe the contributions of this paper are important to the NIPS community. Robust planning is an under-studied problem and the authors address this difficult challenge in a intuitive manner. With that said, I am concerned with the applicability of their formulation beyond toy problems. Their formulation results in an exponential blow up in the size of the problem. In addition, experiments are only performed in toy domains that appear to have been created by the authors specifically for this paper. I presume that with the lack of research into planning in incomplete models, there are no standard domains to run experiments in. However, I am concerned that the approach will be completely infeasible in large domains that arise in the wild. In very large domains, can we ever hope to guarantee a high level of robustness, or are we simply going to be satisfied in just finding a valid plan? How badly do planners such as PFF suffer from such an exponential blowup resulting from the set of all possible complete domain models?

Typos:
- Line 113: "with respects to..."
- Line 135: "Makov Logic Networks"
- Line 430 or 431: "constrasting"

COMMENTS AFTER AUTHOR RESPONSE:

I am glad to hear that proofs will be appearing in a later document. If the Ph.D. proposal will not be accessible soon, I suggest attaching the proofs as supplementary material to the camera-ready paper if this paper is accepted. While I am still somewhat concerned about the applicability beyond toy problems, this is a good first step and future work with heuristics could potentially improve things.
Summary: A good, clearly written paper that addresses an important problem in an intuitive fashion. While it is likely that these ideas are valuable to the planning community, proofs of theorems are absent and it is not clear how applicable the approach is beyond toy problems.

Submitted by Assigned_Reviewer_9

This is an interesting paper that takes on an important weakness of planning methods that makes their application to modern applications difficult - that of having incomplete domain models. The authors model a specific type of incompleteness wherein the designer has some idea of what the unknowns might be, but can't be sure of the exact realized state or other specific quantitative information. This gives us a formulation wherein we look at paths in a probabilistic sense, under the generous execution semantics. This then allows the authors to analyse the problem in terms of model counting, constraint satisfaction and so on, while also allowing for graphical model methods to be used in computation.

Much of this is standard fare these days in areas such as Markov Decision Processes, including the translation into graphical models and solution by inference methods. As the authors note at the very end, the innovation here is in dealing with planning in the sense of operators and factored models.

In my view, I do not believe there is any supplementary material which is where I would have expected to see formal proofs of the theorems. Indeed, the ideas are discussed but it was unclear to me if there was also going to be a formal version.

Lastly, I like the multiple problem types that are formulated. However, I would have expected to see some of these connected to specific implementations in terms of how an MLN or BN exploits the stated formulation in the context of the requirements, e.g., that we are searching for paths above a robustness level. While I see the model formulation at the level it is described, I wonder if there are anymore noteworthy points by the time we are implementing these things in detail, especially given the complexity of the general version of the underlying computational process.
Summary: This is an interesting paper that takes on the problem of incomplete models in a planning paradigm. It would have been interesting to see some more details, but I realize that a lot has already been packed in.

Submitted by Assigned_Reviewer_10

The requirement of domain completeness has indeed become the bottleneck of applications of planning techniques in many real-world domains, such as web service composition, intrusion detection. This paper aims at removing this requirement by constructing robust plans when domains are incomplete. This is a significant contribution.

However, this paper assumes sets of possible preconditions and effects (as well as their weights) are provided as input. This assumption is kind of strong. It would be better if we could see some potential applications (related to this assumption)  in real world application domains.

Did you consider the mutual constraints between single effects and conditional effects in each action when compiling incomplete domain models to CPP models? (I didn't see any explicit description of these constraints.) If not, it is difficult to explain the rationale of the robustness measure since you may take "wrong" domain models into account when doing model counting.

I think an example would be very helpful for readers to understand about the compilation result at the end of Section 4.2.

How many planning problems did you solve for each setting? Does "l/t" mean the "average" length and running time, respectively? 

minors:

page 2: as followed => as follows

page 3, what kind of additional knowledge do we need when modeling possible preconditions/effects using MLNs? 

page 7. "Table 3 and 4" => "Tables 3 and 4"; Captions of "Figure 3" and "Figure 4" should be "Table 3" and "Table 4".

----------------

I would suggest a weak accept (although the problem is interesting, the problem assumption is lack of motivating applications, and questions related to mutual constraints and experimental settings (as far as I am concerned, these questions are important), are unclear to me). 
Summary: The paper breaks new ground in a new direction, but makes too many strong assumptions about the action representation that may or may not hold in the real world. Some examples would help convince readers better.
Author Feedback

Author rebuttal: We thank the reviewers for the helpful comments, and we’re gratified that our work was seen as important and interesting.

We apologize for not attaching the proofs as supplementary materials. This is the first time we submitted to NIPS and we didn’t realize that supplements were allowed. All the proofs are in the first author’s Ph.D. proposal. We include a proof sketch for Theorem 2 at the end of this write-up.

Reviewer 1: The robustness of plans is defined assuming that weights on incompleteness annotations are provided. We envision that, similar to probabilistic modeling (e.g., using Bayesian networks), those weights must be either specified by the domain writers or learned automatically. While we did not discuss this topic in our paper, we think that it is an important research topic. We will clarify this issue in the revised version.

On the question of the performance of classical planners on our problems: We tried to use FF, a popular planner for classical planning (which, in particular, requires complete domains as input), to solve the planning problems in our experiment section when all annotations are ignored. One thing to note is that ignoring possible preconditions or delete effects makes the problems easier, and ignoring add effects makes them harder. Because of this, while FF is at least able to produce valid (if not robust) solutions in Logistics domains, it fails to solve any problems in the Satellite domains (thus, effectively producing plans with robustness zero).

The robustness of plans returned by FF in Logistics domains, however, is always worse than those returned by the compilation approach. Specifically, plans returned in all five variants of the Logistics domains were always 0.3, the failure probability of individual incomplete action schema “load-truck-with-robot-of-Mi”. Since load actions are considered complete, the FF planner uses those related to only one manufacturer, thus does not recognize any opportunity to increase plan robustness. Our compilation approach produces plans with higher robustness, up to (at least) 0.8 probability of success in the largest instance (as shown in Figure 3). It does this by continuously trying load actions performed by multiple types of robots (but with the cost of increasing plan lengths).

Reviewer 2: While we think the compilation approach is a good starting point for this problem, we are also currently working on a heuristic approach to tame the combinatorics. This new approach directly takes the incompleteness annotations into account in the context of a heuristic planner.

Reviewer 3: As we mentioned in the paper, we believe that modeling correlation between incompleteness annotations using Markov Logic networks or Bayesian networks, though interesting from the modeling point of view, in fact requires significant additional effort from the domain writers. We think that it may thus be less helpful in practice. We however note that since Probabilistic-FF already assumes initial belief states to be modeled with Bayesian networks, robust planning with correlated annotations can, in principle, be handled using our compilation approach.

Proof (sketch) for Theorem 2: There is one-to-one mapping between each candidate complete model D_i and a (complete) state s’_{i0} in the initial belief state b_I of the compiled problem. Moreover, if D_i has a probability Pr(D_i) to be the real model, then s’_{i0} also has the same probability in the belief state b_I.

Given our projection over complete model D_i (defined in Section 2), executing plan pi from the state I with respect to D_i results in a sequence of complete state (s_{i1}, ..., s_{i(n+1)}). On the other hand, executing pi' from s’_{i0} in the compiled problem results in a sequence of (singleton) belief states ({ s’_{i1} }, ..., { s’_{i(n+1)} }). By induction we can show that for every proposition p in F and step index j in {0, 1, …, n+1}, p is true in s’_{ij} if and only if it is true in s_{ij}. Therefore, the complete state s_{i(n+1)} achieves goals G if and only if s’_{i(n+1)} achieves G = G'.

Since all actions a’_i in the compiled problem are deterministic and s’_{i0} has probability Pr(D_i) of being in the belief state b_I, the probability that the plan pi' achieves G' is equal to the summation of Pr(D_i) such that s’_{i(n+1)} achieves G, which in turn is equal to the robustness of plan pi (as defined in Equation 2). This proves the theorem.